# Regulation of Protein Structural Changes by Incorporation of a Small-Molecule Linker

**DOI:** 10.3390/ijms19123714

**Published:** 2018-11-22

**Authors:** Youngmin Kim, Cheolhee Yang, Tae Wu Kim, Kamatchi Thamilselvan, Yonggwan Kim, Hyotcherl Ihee

**Affiliations:** 1Center for Nanomaterials and Chemical Reactions, Institute of Basic Science (IBS), Daejeon 34141, Korea; youngmin.kim.k@gmail.com (Y.K.); eangel04@gmail.com (C.Y.); ktwk6660@gmail.com (T.W.K.); nanostar.shruthi@gmail.com (K.T.); anglerlord@gmail.com (Y.K.); 2Department of Chemistry and KI for the BioCentury, Korea Advanced Institute of Science and Technology (KAIST), Daejeon 34141, Korea

**Keywords:** photoactive yellow protein, small-angle X-ray scattering, chemical modification, protein structural dynamics, regulated conformational change

## Abstract

Proteins have the potential to serve as nanomachines with well-controlled structural movements, and artificial control of their conformational changes is highly desirable for successful applications exploiting their dynamic structural characteristics. Here, we demonstrate an experimental approach for regulating the degree of conformational change in proteins by incorporating a small-molecule linker into a well-known photosensitive protein, photoactive yellow protein (PYP), which is sensitized by blue light and undergoes a photo-induced N-terminal protrusion coupled with chromophore-isomerization-triggered conformational changes. Specifically, we introduced thiol groups into specific sites of PYP through site-directed mutagenesis and then covalently conjugated a small-molecule linker into these sites, with the expectation that the linker is likely to constrain the structural changes associated with the attached positions. To investigate the structural dynamics of PYP incorporated with the small-molecule linker (SML-PYP), we employed the combination of small-angle X-ray scattering (SAXS), transient absorption (TA) spectroscopy and experiment-restrained rigid-body molecular dynamics (MD) simulation. Our results show that SML-PYP exhibits much reduced structural changes during photo-induced signaling as compared to wild-type PYP. This demonstrates that incorporating an external molecular linker can limit photo-induced structural dynamics of the protein and may be used as a strategy for fine control of protein structural dynamics in nanomachines.

## 1. Introduction

Various nanomachines based on small-molecule moieties have been designed and synthesized to control structural changes and mimic numerous mechanical movements at atomic-length scales [1,2]. However, the nanomachines composed of small organic compounds may not be suitable for applications in living organisms because of their potential toxicity. In this regard, nanomachines composed of proteins having low toxicity have a potential to serve as a good alternative. However, controlling the structural changes of proteins and manipulating motion via structural regulation of proteins in molecular nanomachines are challenging, since proteins are complicated macromolecules with many levels of structural organization from the primary structure (a sequence of amino acid residues) to the quaternary structure (interaction of protein subunits). In this regard, one of the important challenges in developing protein-based nanomachines is to achieve fine control of the mechanical motions within proteins.

It has been reported that the structural changes of photo-responsive organic molecules bound to photo-insensitive proteins can induce subsequent structural changes in the protein. In this way, photo-insensitive proteins can be turned into photoactive proteins. A well-known example is the control of protein structural changes using azo-benzene derivatives [3]. In the case of a protein nanocage [4], an ATP-driven chaperonin was turned into a light-driven cage by binding the azobenzene-based linkers. These studies have focused on inducing structural changes of proteins using photo-switchable small molecules. In this work, we investigated a reverse version of this approach. In other words, we chose a protein known to undergo a large conformational change and attached a non-photoactive small-molecule linker to the protein to limit the extensive structural change. We aimed at restricting pre-existing structural changes or scaling down structural changes to achieve a fine control of structural changes, rather than creating new structural changes. Particularly, we investigated the possibility and the mechanism of regulating the degree of conformational change by mutations and intramolecular conjugations using a linker molecule.

For this purpose, we used photoactive yellow protein (PYP). PYP, which has *p*-coumaric acid (*p*CA) as the chromophore, is an attractive model system for understanding photoreception and the subsequent signal transduction process responsible for the negative phototaxis in *Halorhodospira halophila* [5]. Upon the absorption of blue light, PYP undergoes a volume-conserving trans-cis isomerization of the *p*CA chromophore, which leads to a significant conformational change of the protein. This photoreaction of PYP is accompanied by changes in the absorption spectra, refractive indices, ion complexation, and electrochemical properties [6,7,8,9,10,11,12]. The ground state of wild-type PYP, known as pG, has a globular protein conformation with the trans configuration of *p*CA chromophore, whereas a light-activated state known as pB (or pB_2_) or I_2_ has a relatively expanded conformation including the protrusion of the N-terminus and the cis configuration of the chromophore. The photoreaction of PYP involves the formation of the light-activated state populated in the millisecond time region and the relaxation of the light-activated state to the ground state takes place within milliseconds to minutes. Since sensitization with blue light in PYP causes a global conformational change and such a photoreaction is photo-reversible, we used PYP as a target system in order to demonstrate the new approach for regulating global structural changes in proteins.

We employed the incorporation of a chemical reagent, acting as a bridge connecting the protein moieties, to the modified PYP. To introduce a small-molecule linker into PYP, two amino acid residues of wild-type PYP were mutated into cysteine residues with thiol group side chains. Two thiol groups were covalently conjugated with the ends of a thiol-specific cross-linker. When PYP absorbs blue light, the cross-linker would prevent the protrusion of the N-terminus and thus the structural change of PYP can be restricted as shown in Figure 1.

To investigate the effect of the molecular linker on the N-terminus of PYP with respect to the protein conformation, we used small-angle X-ray scattering (SAXS) combined with systematic structural analysis based on experiment-restrained rigid-body molecular dynamics (MD) simulation to determine the protein shape, size, and conformational differences between the states with and without blue light irradiation and transient absorption (TA) spectroscopy to analyze the kinetics. We analyzed the conformational changes of PYP induced by conjugation of the cross-linker and the conformational changes of the modified PYP induced by irradiation. 

## 2. Results

### 2.1. Sample Characterization

We considered the following points when we chose the two positions at which cysteine mutations could be introduced to conjugate the small linker molecule. Since our goal is to restrict the photoinduced N-terminus movement of PYP, one of the positions should be located at the N-terminus. For this position, we chose the 7th (glycine) and 13th (asparagine) positions considering the minimal interference with other residues and the side chain direction pointing toward the protein body. For the other position in the protein body, we chose the 91st (methionine) and 113th (leucine) positions considering the minimal interference and the side chain direction pointing toward the N-terminus. Another consideration at this point was that the selected position at the N-terminus and the position at the protein body should be separated by a distance similar to or shorter than the length of the small-molecule linker. We generated and tested various cysteine-mutated PYPs with the selected mutation positions and found that only the mutant residues at the 7th (glycine) and 91st (methionine) positions were successfully conjugated with the 1,8-bismaleimidodiethyleneglycol (BM(PEG)_2_) linker (see Appendix A). Although the length (~14 Å) of BM(PEG)_2_ is longer than the distance (~9 Å) between the 7th and 91st residues, the small molecule, which is flexible, was successfully conjugated between the mutated positions. This PYP conjugated with the BM(PEG)_2_ linker was named SML-PYP, where SML stands for small-molecule linker. After incorporating the cross-linker to the PYP mutants, the samples were analyzed using UV-vis absorption spectroscopy and matrix-assisted laser desorption/ionizaiton time-of-flight (MALDI/TOF) spectrometry. The UV-vis absorption spectrum of SML-PYP is almost the same as the previously reported absorption spectrum of wild-type PYP (Appendix A), indicating that the environment around the chromophore binding pocket is not affected by the double mutation and the incorporation of the linker molecule. Based on the results from the measurement of MALDI/TOF, we confirmed that the molecular weight of SML-PYP is equal to the sum of the molecular weights of the wild-type PYP and the linker molecule (Appendix A). Moreover, Thiol Fluorescent Detection Kit (Invitrogen™) data showed almost no free thiols in the sample solution of SML-PYP (Appendix A), implying that most of the side chains of cysteine in the PYP mutant formed a thioether bond with the linker. We also measured the CD (circular dichroism) spectrum of SML-PYP to check the structural stability (Appendix A).

### 2.2. Transient Absorption (TA) Spectroscopy 

From the fitting result of TA data, four-time constants, τ_1_, τ_2_, τ_3_, and τ_4_, were determined (Table 1). TA data of wild-type PYP and SML-PYP exhibited similar spectra and kinetics (Figure 2 and Table 1), suggesting that the conjugation of the linker and the double mutation of PYP do not affect its overall kinetics. The photocycle of PYP has been explained by two kinetic models, which are a sequential model (Figure 2d) [13,14,15] and a parallel model (Figure 2e) [11,16,17]. In the photocycle of PYP, two red-shifted intermediates (pR_1_ and pR_2_) and two blue-shifted intermediates (pB_1_ and pB_2_) compared to the spectrum of the ground state (pG) are formed after blue light irradiation. Based on the sequential model, τ_1_ and τ_2_ correspond to the transition from pR_1_ to pR_2_ and from pR_2_ to pB1, respectively. Based on the parallel model, τ_1_ and τ_2_ correspond to the transition from pR_1_ to pB_1_ and from pR_2_ to pB_1_, respectively. In both kinetic models, τ_3_ and τ_4_ can be assigned as the transition from pB_1_ to pB_2_ and dark recovery from pB_2_ state to pG state, respectively. Unlike other transitions, the formation of pB_2_ (τ_3_) of SML-PYP is almost two times faster than that of wild-type PYP (Table 1). The pB_2_ state of wild-type PYP is a putative signaling state of PYP and shows the largest structural changes compared to the pG state during the photocycle of PYP. Hence, the accelerated formation of pB_2_ of SML-PYP might indicate that the structural changes of SML-PYP are regulated by the cross-linker. 

### 2.3. Analysis of SAXS Data for the Ground and the Light-Activated States of SML-PYP

To investigate the structure of the ground and the light-activated states of SML-PYP, we performed the small-angle X-ray scattering (SAXS) experiment with and without continuous blue light irradiation (Figure 3a,b). PYP is a highly soluble protein [5] and even after the incorporation of BM(PEG)_2_ the solubility of the protein was maintained. To rule out the distortion of X-ray scattering curve originated from the aggregated species, the solution was cautiously centrifuged at 10,000 *g* for 10 min prior to performing the SAXS measurements. As shown in Appendix A, the Guinier plot of the original SAXS data shows an excellent linearity, which implies that the scattering signal in the small-angle region is free from aggregation. Radius of gyration (*R*g) values for the dark and light states of SML-PYP were determined as 15.3 and 15.4 Å, respectively, as determined with the PRIMUS program of the ATSAS software package [18,19,20]. We assumed full conversion from dark state to light state because we estimated the conversion rate to be over 97.5% based on the UV-vis spectrum under continuous light-emitting diode (LED) illumination (see Appendix A). The *R*g value of the ground state for SML-PYP is similar to that of the light-activated state for wild-type PYP (15.6 Å) rather than that of the dark state for wild-type PYP. Since the size of the small-molecule linker itself is too small to affect the *R*g value, this increased *R*g value might be due to the conformational change of the protein body, and suggests that, compared to that of wild-type PYP, the ground state of SML-PYP has a more swollen structure, which possibly has the N-terminal protrusion due to the instability arising from the linker conjugation. The *R*g value of light-activated state for SML-PYP is slightly larger than that of the ground state for SML-PYP. The difference between the *R*g values of the light and ground states of SML-PYP is only 0.13 Å, which is much smaller than that of the wild-type PYP (1.2 Å), indicating that the structural change upon light irradiation of SML-PYP is indeed restricted compared to that of wild-type PYP, fulfilling our goal of regulating the degree of structural change by small-molecule linkage. 

### 2.4. Molecular Shape Reconstruction from the SAXS Data

Prior to shape reconstruction, the pair distribution functions, *P(r)*, of the dark and light states for SML-PYP were obtained using the GNOM program (Figure 3c) [21]. *P(r)* of the dark state shows the main distribution at ~20 Å and a minor distribution reaching 50 Å. *P(r)* of the light state shows a similar main distribution with slightly less population compared to that of the dark state and a minor distribution reaching 53 Å. The ratio between the main and minor distributions slightly changed between the light and dark states, indicating that blue light irradiation resulted in restricted structural changes of SML-PYP, probably because of slight unfolding.

Next, we performed shape reconstruction using the DAMMIF program and GNOM data to estimate the shapes of the dark and light states of SML-PYP (Appendix A) [22]. The reconstructed shapes of the dark and light states of SML-PYP were obtained for each state (Figure 4a–c). The light state structure is superimposed on the dark state structure of SML-PYP in Figure 4c. The light state shape of SML-PYP is slightly more anisotropic and elongated along the long axis compared to the dark state shape, but the degree of change is small.

### 2.5. Experiment-Restrained Rigid-Body MD Simulation 

To more precisely define which part of the protein is responsible for the difference between SML-PYP and wild-type PYP, we also performed the experiment-restrained rigid-body MD simulations and thus determined the structure of the dark and light states (Appendix A and Figure 4d–f) [7,23]. In this simulation, we used a q-value as large as 0.5 Å^-1^ and produced a model based on all available data to facilitate higher resolution of the protein shape. As shown in Figure 4d, the dark state structure of SML-PYP shows the protrusion of the N-terminus, as expected in the comparison of *R*g values and reconstructed shapes between SML-PYP and wild-type PYP (Figure 4g). Moreover, the light state structure of SML-PYP shows slight elongation along the long axis compared to the dark state, similar to the result obtained with the shape reconstruction (Figure 4h). The degree of the elongation due to the dark to light conversion upon blue light irradiation for SML-PYP is not as significant as that for wild-type PYP, which is consistent with the relatively small *R*g value changes of SML-PYP. The mutated positions of the 7th and 91st residues are located on the N-terminus and core, respectively, and the intramolecular conjugation using a linker molecule probably constrains the N-terminus structural changes of SML-PYP compared to those of the wild-type PYP. This scenario is schematically represented in Figure 5.

## 3. Discussion

It is important to understand the mechanical and structural changes in proteins to achieve fine control of structural changes in protein-based nanomachines [24]. To do so, we investigated a new method which is the incorporation of a small-molecule linker into the protein. This work may serve as the starting point for the development of strategies to regulate protein structural changes.

The CD spectrum of the mutant PYP (G7C-M91C PYP) is quite similar to that of wild-type PYP, whereas the CD spectrum of SML-PYP indicates that its α-helix content is moderately reduced (Appendix A). The SAXS data show an increase of *R*g value in the ground state (pG) of SML-PYP by 0.9 Å compared with that of wild-type PYP, which corresponds to an approximate volume increase of 5500 Å^3^. The volume of the six residues in the N-terminal region would be about 147 Å^3^ plus the volume of the hydration shell and thus this alone cannot account for the 0.9 Å increase in *R*g value. On the other hand, the environment around the chromophore region of SML-PYP is expected to be well conserved and similar to that of the wild-type based on the similarity of the UV-vis spectra (see Appendix A). The reduced α-helix content, the increased *R*g value and the well-conserved chromophore environment all together suggest that SML-PYP has a swollen conformation with a well-conserved core. In accordance with this possibility, the molecular shape reconstructed from the SAXS data shows a partial protrusion in the ground state of SML-PYP (Figure 4a). The molecular origin of the partial protrusion can be identified by comparing the shape and the structure from the experiment-restrained rigid-body MD simulation (Figure 4d). The protrusion of the reconstructed shape well matches the N-terminal region in the structure from the experiment-restrained rigid-body MD simulation, confirming that the N-terminal region is indeed responsible for the protrusion. This result, which we did not anticipate when we had chosen the cross-linking positions and the small-molecule linker, shows that the overall shape of the protein molecule can be easily affected by the small-molecule linker. We note that, because the experiment-restrained rigid-body MD simulation yields only a single protein structure, the resulting structure cannot properly represent the ensemble with the conformational heterogeneity. Nonetheless, judging from the good agreement between the dummy-atom modeled shape and the structure from the experiment-restrained rigid-body MD simulation shown in Figure 4, we suggest that the latter may represent a protein conformation of the microstate included in the ensemble.

A study using time-resolved X-ray solution scattering reports large changes in the *R*g value from 14.4 to 15.6 Å for wild-type PYP upon blue light irradiation [11]. However, in the case of SML-PYP, the *R*g value changed from 15.3 to 15.5 Å upon blue light irradiation. The structure of the light state determined by the shape reconstruction and the experiment-restrained rigid-body MD simulation also shows only minor N-terminal structural changes including a slight elongation of the protein along the long axis, whereas wild-type PYP shows dramatic structural changes accompanied by N-terminal protrusion. The reduced structural changes induced by the small-molecule linker demonstrate that our designed strategy works. The marginal change of N-terminal protrusion of SML-PYP may originate from the cross-linking incorporated in PYP that significantly restricts the mechanical motion of the N-terminus. Thus, we demonstrate that the structural changes in a protein can be restricted by introducing a small-molecule linker (Figure 5 and Appendix A). Furthermore, this result is additional evidence that the structural changes of the N-terminus are essential for the formation of the light state of wild-type PYP because our results confirm cross-linking the N-terminus and the core of PYP restricts the photoinduced protrusion of the N-terminal region. In other words, this result confirms the suggestion by previous studies [11,12] that the N-terminus protrusion is the major structural change in the late intermediate stage. 

The TA results show that mutation and cross-linking affect the kinetics of the formation of the light state, pB_2_, in SML-PYP. The time constant of the pB_2_ formation for SML-PYP is two times faster (480 µs) than that for wild-type PYP (920 μs). This can be rationalized by considering the structural aspect of SML-PYP, i.e. the pG structure of SML-PYP is a swollen conformation relative to that of wild-type. This pre-swollen conformation in pG may facilitate the conversion to pB_2_, resulting in the modest acceleration of the pB_1_ → pB_2_ transition.

## 4. Materials and Methods

### 4.1. Preparation of PYP with Blocked N-terminal Movement

We tried to prepare various constructs of PYP, with different mutated pairs of amino acid residues: 7th and 91st, 13th and 113th, and 7th and 113th residues. The pair of amino acids were mutated into cysteines, with a thiol (-SH) side chain. The mutants were tested for the potential of the thiol groups of the cysteines to conjugate with various linkers, such as bismaleimidoethane (BMOE), 1,8-bismaleimidodiethyleneglycol (BM(PEG)_2_), and 1,4-bismaleimidobutane (BMB). The modified PYP was purified using previously reported methods (see Appendix A) [25]. After incubating with the linker, the PYPs were analyzed by MALDI/TOF and Thiol Fluorescent Detection Kit (Invitrogen™) to confirm the degree of conjugation. CD spectra were also measured to check structural stability (Appendix A).

### 4.2. Transient Absorption (TA) Measurement

TA measurement was performed by following previously reported procedures [26,27,28,29]. A pump beam (460 nm) was produced by an optical parametric oscillator (LT-2214-PC, LOTIS, Minsk, Belarus) pumped by a third harmonics (355 nm) of an Nd:YAG laser (NL 301G, EKSPLA, Vilnius, Lithuania). A probe beam was generated by a continuous wave 250 W Xe-lamp (66902, Newport, Irvine, CA, USA). The probe light and the pump laser beam were intersected on a sample position at an acute angle to each other to maximize the overlap area. TA signal was detected by a combination of a spectrometer (SpectraPro 2300i, Princeton Instrument, Acton, MA, USA) and a detector. We used two types of detectors: an ICCD (iStar, Andor, Belfast, United Kingdom) was used for TA spectra measurements and a photomultiplier tube (PMT) connected with a digital oscilloscope (TDS 3052B, Tektronix, Beaverton, OR, USA) was used for temporal profile measurements. To improve the signal-to-noise ratio, the TA spectra and the temporal profiles were averaged 200 times and 320 times, respectively. The concentrations of PYPs were adjusted to be about 70 μM. The sample solution was injected and flowed by using a syringe pump and a 2 mm quartz flow cell. The repetition rate of the pump beam was 1 or 0.5 Hz.

### 4.3. SAXS

X-ray scattering data as a function of scattering vector (*q*) were collected at the 4C SAXS II beamline at Pohang Light Source (PLS-II) in Korea. Monochromatic X-ray pulse with the energy of 16.90 keV was used for data collection. Static X-ray scattering patterns were recorded using a MarCCD detector (Rayonix, Evanston, IL, USA). The sample-to-detector distance was set at 1 m. The *q* ranges of the data for dark and light states were 0.03–0.50 Å^−1^ and 0.03–0.43 Å^−1^, respectively. The protein was dissolved in the buffer solution of 20 mM Tris (pH 7.0) with 20 mM NaCl and the concentration of the protein was 10 mg/mL. The sample solution was centrifuged at 10,000 *g* for 10 min to remove unwanted aggregated particles and only the supernatant was used for the measurement prior to performing the SAXS measurements. To populate the light state, the protein sample was continuously irradiated by focused blue LED light during the measurement. To avoid any damages induced from X-ray and optical lights, the X-ray scattering curves of the SML-PYP were obtained by employing a flow cell system (Appendix A). The PYP sample solution was injected into a capillary at 100 μL/min. The capillary was 1.0 mm thick, and the X-ray exposure time was 60 s. To confirm photo-conversion ratio to pB_2_, we measured a UV-vis spectrum using the same setup (Appendix A).

### 4.4. SAXS Analysis and Molecular Shape Reconstruction

The SAXS data were processed and analyzed using the ATSAS package (http://www.embl-hamburg.de/biosaxs/software.html). One-dimensional scattering data, *I(q)*, were obtained by azimuthal averaging. Scattering intensities for the pure buffer solution were measured by a separate experiment to determine the background intensity, and the *I(q)* were extrapolated to *q* = 0 by GNOM program. The *R*g value and the pair distribution function, *P(r)*, were calculated through indirect Fourier transformation [21]. The q and real space range for the dark state of SML-PYP are 0.05–0.50 Å^−1^ and 0.00–50.06 Å, respectively, and those for the light state of SML-PYP are 0.05–0.40 Å^−1^ and 0.00–54.16 Å, respectively. The maximum dimension (*D*max), which is the maximum distance between scatterers was determined from PRIMUS at ATSAS. Using DAMMIF program [22], ab-initio shape reconstruction for the X-ray data with the q range of 0.001–0.29 Å^−1^ and the optimal *P(r)* function was performed to reconstruct the shape of the protein. This shape reconstruction was repeated nineteen times for each data. Representative structures were generated by superimposing each structure determined from DAMMIF process result onto a template structure using DAMSEL and DAMSUP. Nineteen low-resolution dummy atomic models were averaged using DAMAVER [30]. DAMFILT was used to obtain a representative structure with high-probability densities. The reconstructed shape was compared with those of models obtained from the experiment-restrained rigid-body MD simulations to determine the conformational changes induced by blue light irradiation at the atomic level. The reconstructed shapes and experiment-restrained rigid-body MD simulation results were superimposed using SUPCOMB [31].

### 4.5. SAXS Curve Analysis with the Experiment-Restrained Rigid-Body MD Simulation

Structures and shape of SML-PYP in the dark and light state were obtained from the experiment-restrained rigid-body MD simulations (see Appendix A). We used the crystal structure of PYP (PDB ID: 2PHY) and manually attached it with the molecular linker for use as the initial structure in the experiment-restrained rigid-body MD simulation. In this simulation, 2–3 amino acids in the loop region and 4–12 amino acids in the alpha helix and beta sheet were grouped as a single rigid body to reduce the number of parameters and calculation time for the simulation. In total, ~29 rigid bodies were used in the experiment-restrained rigid-body MD simulations. The CRYSOL program [32] was used to calculate the theoretical X-ray scattering curve of the refined structure in the experiment-restrained rigid-body MD simulations. In this simulation, we minimized the value of a target function that represents the discrepancy between the GNOM curves originated from experiment and the theoretical scattering curves on the q range of 0–0.43 Å^−1^. In the target function, the van der Waals forces between rigid bodies and the correction terms for bond lengths were also considered to minimize chemical instability. Then, we fitted the experimental curve by changing the CRYSOL parameters, such as the contrast in the hydration shell, the radius of the atomic group, excluded volume, and scaling factor between the experimental and theoretical results. For the fitting, we employed the MINUIT program in ROOT [33] from CERN. The experiment-restrained rigid-body MD simulation and the fit of the CRYSOL parameter were performed iteratively until the smallest chi-square value was achieved. Finally, we determined the best model exhibiting a good agreement between the experimental SAXS curve and the theoretical scattering curve. 

## 5. Conclusions

SAXS and TA results show that cross-linking between the N-terminus and the core of PYP with BM(PEG)_2_ may constrain the N-terminal movement of PYP. SML-PYP slightly increased in size, including elongation along the long axis upon blue light irradiation, whereas wild-type PYP exhibited large structural changes accompanied by N-terminal protrusion. We demonstrate that the incorporation of a small-molecule linker can be used for regulating the structural dynamics of the protein. The ability to regulate the mechanical motions with cross-linking may provide a new strategy for the mechanical manipulation of protein nanotechnology and lead to novel nanomachines for a variety of applications. 

## Figures and Tables

**Figure 1 ijms-19-03714-f001:**
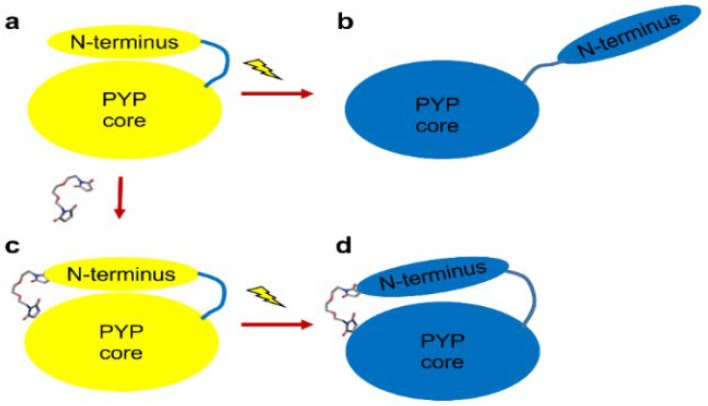
Schematic representation of the expected conformational change in the chemically modified photoactive yellow protein (PYP). The chemically modified protein includes the small-molecule linker (stick) covalently conjugated with the thiol groups (red-color) at the 7th and 91st residues. Wild-type PYP undergoes an N-terminal protrusion upon blue light excitation (**a** → **b**). In the chemically modified PYP (SML-PYP), the small-molecule linker plays a critical role in regulating the degree of structural change in the photoreaction of PYP, thereby limiting the N-terminus protrusion (**c** → **d**).

**Figure 2 ijms-19-03714-f002:**
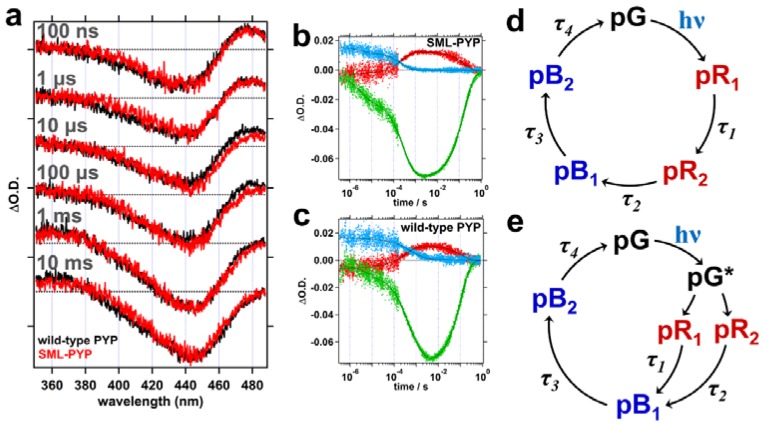
(**a**) Transient Absorption (TA) spectra of SML-PYP (red) are compared with those of wild-type PYP (black). Time delays for the TA spectra are (from top to bottom): 100 ns, 1 μs, 10 μs, 100 μs, 1 ms, and 10 ms. ΔO.D. stands for the optical density change and the spectra are vertically offset for clear presentation. (**b**,**c**) Temporal profiles of TA signal for (**b**) SML-PYP and (**c**) wild-type PYP measured at the probe wavelengths of 380 (red), 465 (green), and 494 nm (blue) are shown. The decay profiles of each protein were globally fit by the sum of exponential functions consisting of four exponentials and an offset. The fit lines are shown in black dotted lines. The time constants determined by the fit are listed in Table 1. (**d**,**e**) Schematic illustrations of (**d**) the sequential kinetic model and (**e**) the parallel kinetic model for the photocycle of PYP. hν indicates the optical excitation.

**Figure 3 ijms-19-03714-f003:**
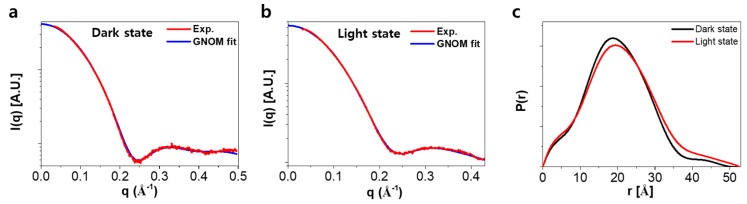
SAXS (red) and GNOM fit results (blue) for BM(PEG)_2_-incorporated PYP (SML-PYP) in the dark (**a**) and the light (**b**) states. (**c**) Pair distribution, *P(r)*, of SML-PYP with (red) and without (black) exposure to light. *P(r)* functions show that the light state of SML-PYP has a larger *R*max value. A. U. stands for arbitrary unit.

**Figure 4 ijms-19-03714-f004:**
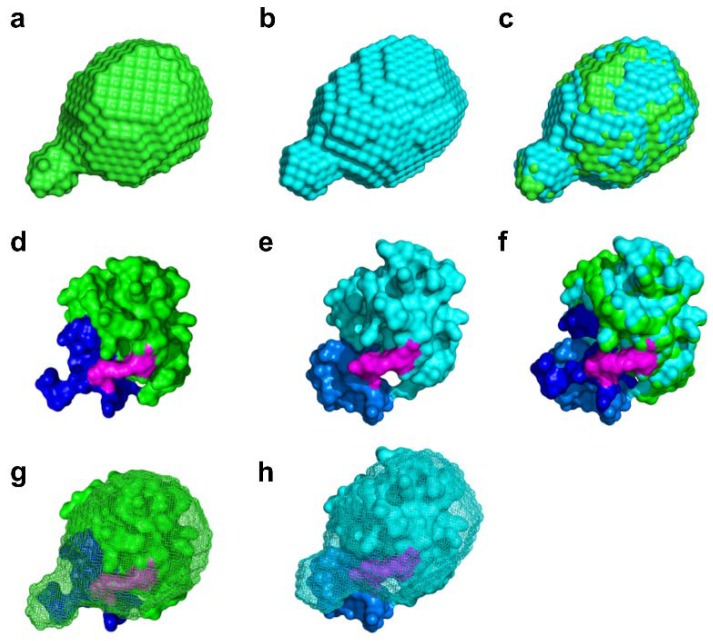
Molecular structures modeled from the low-resolution shape reconstruction and the experiment-restrained rigid-body MD simulation. (**a**–**c**) Low-resolution shape reconstruction of SML-PYP in the dark (**a**) and light (**b**) states, and the superimposed image (**c**) of DAMMIF filtered results. (**d**–**f**) Structural representation of SML-PYP in the dark (**d**) and light (**e**) states from the experiment-restrained rigid-body MD simulation, and the resulting superimposed image (**f**). (**g**,**h**) Comparison of structures from the low-resolution shape reconstruction (mesh) and the experiment-restrained rigid-body MD simulation (sphere-dot) of SML-PYP in the dark (**g**) and light (**h**) states. Blue and marine colors indicate 27 amino acids in the N-terminal region of SML-PYP in the dark and light states, respectively. Green and cyan colors indicate core region of SML-PYP in the dark and light states. Magenta color indicates the BM(PEG)_2_ molecule.

**Figure 5 ijms-19-03714-f005:**
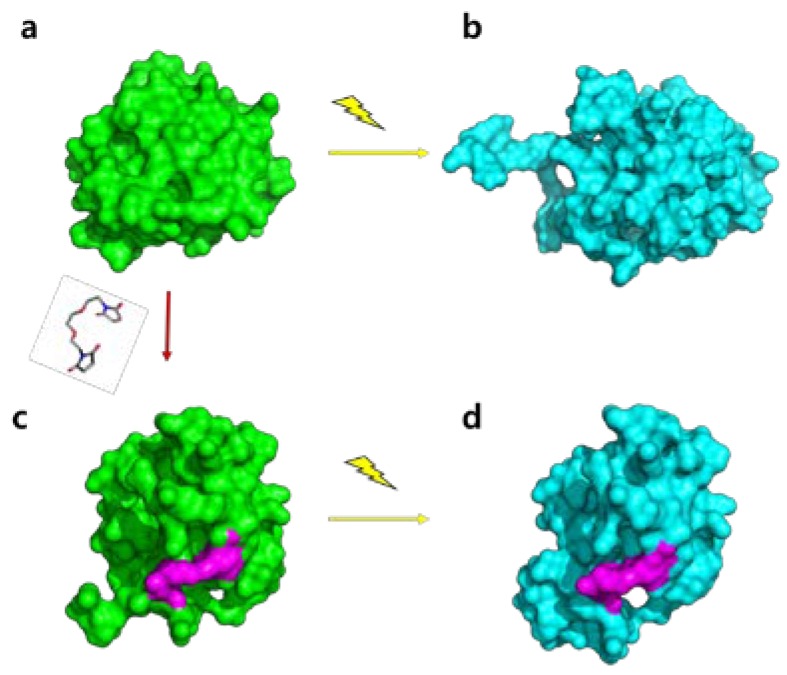
Schematic illustration of the structural changes in PYP induced by small-molecule binding (**a** → **c**) and blue light irradiation (**a** → **b**, **c** → **d**). (**a**) The dark state of wild-type PYP. (**b**) The light state of wild-type PYP. (**c**) The dark state of SML-PYP. (**d**) The light state of SML-PYP. The small-molecule linker is shown in magenta. Wild-type PYP dark and light structures are from PDB (protein data bank) ID, 2PHY and 2KX6, respectively, and the dark and light structures of SML-PYP are from this work.

**Table 1 ijms-19-03714-t001:** Time constants of SML-PYP and wild-type PYP determined by the fitting of TA data.

Fitting Results	SML-PYP	Wild-Type PYP
τ1 (μs)	3.3 ± 1.3	4.4 ± 1.2
τ2 (μs)	200 ± 51	210 ± 49
τ3 (μs)	480 ± 140	920 ± 260
τ4 (ms)	170 ± 3	120 ± 10

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
