# Peer review of "Regulation of Protein Structural Changes by Incorporation of a Small-Molecule Linker"

_ijms, 2018, doi:10.3390/ijms19123714_

Round 1
Reviewer 1 Report
In this paper, authors propose the design of protein manipulation by using small molecule linker (SML). This approach is canonical, but the application has been limited. Authors succeeded to regulate the conformational change and kinetics of photoactive yellow protein. Although the effect was different from that expected before the experiments, these findings provide a suggestion in the future research. The experiments were designed and conducted by experienced research group. However, the interpretation and discussion do not appear to be enough for the readers who are interested in molecular sciences. More detailed discussion in amino acid level would be desired. I understand that only the low resolution analysis was carried out in the present study, but detailed structural information is accumulated and available in PYP.
Major Points
(1) In general, continuous light illumination of photocyclic pigment cannot perfectly convert the ground state to intermediate state. Thus I guess that the authors would estimate the fraction of pG state under illumination, and contribution of pG was subtracted to obtain Rg of pB. Authors should state the estimation of the amount of pG, because accuracy of Rg value depends on it.
(2) It is well known that SAXS is sensitive to aggregation of the protein in the sample. The possible contribution of aggregation to the increased Rg of SML-PYP in pG state should be discussed.
(3) If the effect of aggregation is excluded, SML increases the Rg of pG state by 0.9 A. Because SML fixes residues 7 and 91, protrusion of N-terminus of SNL-PYP is likely to be composed of only 6 amino acid residues. Is it consistent with the difference in Rg (0.9 A) between pG states of WT and SML-PYP?
(4) Deletion of N-terminal 6 amino acid residues results in the substantially slow recovery of dark state (Harigai et al., J. Biochem. 2001). If SML detaches the N-terminal 6 amino acid residues, it is likely that the recovery of pG is much slower than the experimental results. Are there any consistent explanation?
(5) Comparison of pG structures between WT PYP and SML-PYP, and that of pB2 structures between WT PYP and SML-PYP would be informative.
(6) Authors attributed the fast pB1-pB2 transition of SML-PYP to easy approach of proton from outside water to chromophore caused by SML. I do not accept this idea because absorption spectrum of pB1 strongly suggests that the chromophore of pB1 is already protonated. Are there any evidence that the chromophore receive proton from outside water rather than Glu46?
Minor points:
(1) Line 45, photorespomsive -> photoresponsive
(2) In the legend of Figure 1, it is indicated that the thiol groups are shown in red (line 83), but there are no red residues in Figure 1. The clear presentation of the position of closslink using crystal structure is helpful for readers.
(3) Line 93, limitedd -> limited
(4) Line 233, S,ML -> SML
(5) The x-axis label of Figure S1 should be Wavelength. For easy comparison, the spectrum of WT should be overlapped.
(6) The list of reference is missing in Supporting Information.
Author Response
Response to Reviewer 1 Comments
Comments and Suggestions for Authors:
In this paper, authors propose the design of protein manipulation by using small molecule linker (SML). This approach is canonical, but the application has been limited. Authors succeeded to regulate the conformational change and kinetics of photoactive yellow protein. Although the effect was different from that expected before the experiments, these findings provide a suggestion in the future research. The experiments were designed and conducted by experienced research group. However, the interpretation and discussion do not appear to be enough for the readers who are interested in molecular sciences. More detailed discussion in amino acid level would be desired. I understand that only the low resolution analysis was carried out in the present study, but detailed structural information is accumulated and available in PYP.
Response 1:To address the reviewer’s comment, we have added a new figure (Figure S7) in the Supplementary Material. Figure S7 shows the UV-vis spectra of dark-adapted and light-illuminated PYP. The spectrum measured by the continuous LED illumination clearly shows an increase in the new absorption feature in the spectral region below 400 nm as well as a decrease of the lowest absorption band with the maximum at 446 nm. This spectral change indicates that under light illumination, the light-activated state, expected as pB state, is formed in concurrence with the depletion of the pG state. UV-vis spectra showed that the 446 nm absorbance 0f 0.40 prior to continuous LED illumination was greatly reduced to 0.01 upon continuous LED illumination, and thus the fraction of unexcited species in the light illumination is estimated to be less than 2.5%. When this spectrum was measured, the concentration of the solution was 10 µM and the volume was 2 mL. For the SAXS measurement, the concentration was 0.71 mM, but the volume (less than 1 µL) was much smaller (10 µM/2 mL correspond to 0.71 mM/28 µL) so that the overall number of proteins in the SAXS measurement is even smaller (1uL : 28µL) than that in the UV-vis measurement. Even though the concentration of protein solution used for the spectroscopic (10 µM) and the X-ray scattering (710 µM) measurements were different from each other, we used the same LED source with tighter focusing in the SAXS experiment to further increase the number of photons relative to that in the spectroscopic measurement. Thus the fraction of pG under continuous LED illumination is expected be even much smaller than 2.5% in the SAXS measurement.
(Figure S7 is in the attachment.)
Figure S7. UV-vis spectra of PYP with/without continuous LED illumination. The spectrum with continuous LED illumination barely shows the peak corresponding to pG, indicating that pG has been converted to pB under the used illumination condition. The fraction of pG is estimated to be less than 2.5%. When this spectrum was measured, the concentration of the solution was 10 uM and the volume was 2 mL. For the SAXS measurement, the concentration was 0.71 mM, but the volume (1 uL) was much smaller so that the overall number of proteins in the SAXS measurement is even smaller than that in the UV-vis measurement. In addition, the same LED source was used with tighter focusing for the SAXS measurement and thus the pG fraction under continuous LED illumination is expected be even much smaller than 2.5% in the SAXS measurement.
In addition, we have added the following sentence in the discussion section of the main text.
“We assumed full conversion from dark state to light state because we estimated the conversion rate over 97.5% based on the UV-vis spectrum under continuous LED illumination (see Figure S7).”
Response 2: To address the reviewer’s concern, we have added the following discussion in the main text’s result.
“PYP is a highly soluble protein [5] and even after the incorporation of BM(PEG)2 the solubility of the protein was maintained. To rule out the distortion of X-ray scattering curve originated from the aggregated species, the solution was cautiously centrifuged at 10,000×g for 10 min prior to performing the SAXS measurements. As shown in Figure S6, the Guinier plot of the original SAXS data shows an excellent linearity, which implies that the scattering signal in the small-angle region is free from aggregation.”
In addition, we added a new figure (Figure S6) and a brief discussion as follows.
(Figure S6 is in the attachment)
Figure S6. Guinier region of SAXS data for BM(PEG)2-incorporated PYP (SML-PYP) in the dark state. PYP is a highly soluble protein and even after the incorporation of BM(PEG)2 the solubility of the protein was maintained. To avoid any distortion of the X-ray scattering curve originated from the aggregated species, the solution was cautiously centrifuged at 10,000×g for 10 min prior to performing the SAXS measurements. The Guinier plot of the original SAXS data shows excellent linearity with R-square value of 0.9999, confirming no aggregation in the protein solution.
Response 3: The volume of 6 residues would be about 147 Å3 plus the volume of the hydration shell and thus this alone cannot account for the increase of Rg value by 0.9 Å, which corresponds to an approximate volume increase of 5,500 Å3. Therefore the linkage of BM(PEG)2 seems to destabilize the protein so that the overall volume of the protein is increased (Especially, the N-terminal region at 1~28 amino acid).
In addition, we have added the following sentence in the discussion section of the main text.
“The SAXS data show the increased of Rg value in SML-PYP by 0.9 Å, which corresponds to an approximate volume increase of 5,500 Å3. The volume of 6 residues in the N-terminal region would be about 147 Å3 plus the volume of the hydration shell and thus this alone cannot account for the increase of Rg value by 0.9 Å.”
Response 4: The authors of the paper suggested that in wild-type PYP, 6 residues at the N-terminal interact with the core of the protein during pG recovery, but the deletion removes such interaction, thereby slowing recovery down. In SML-PYP, the 6 residues in the N-terminal region is not deleted but still attached to the protein body, and thus such an interaction would be still facilitated (also have cross-linkage between 7th and 91st). In addition, we note that the of Rg value of pG of SML-PYP is already quite similar to that of pB of SML-PYP.
Response 5: Following the reviewer’s suggestion, we have added a new figure (Figure S10), which compares the pG structures between WT PYP and SML-PYP and also pB between WT PYP and SML-PYP.
(Figure S10 is in the attachment)
Figure S10. Protein structures refined from the experiment-restrained rigid-body MD simulation and previous reported wild-type PYP and pB2 structure. (a) Comparison of pG structures of SML-PYP and wild-type PYP. The protein conformations color-coded by green and gray colors correspond to pG states of SML-PYP and wild-type PYP, respectively. (b) Comparison of SML-PYP and pB2 structures of wild-type (PDB ID: 2KX6). Blue and gray colors indicate pB2 of SML-PYP and wild-type PYP, respectively. The magenta mesh shows the BM(PEG)2 moiety covalently linked to the 7th and 91st residues in the protein.
Response 6: We totally agree with this comment. Following the reviewer’s suggestion, we modified the discussion section in the main text as follows.
“This can be rationalized by considering the structural aspect of SML-PYP, that is the pG structure of SML-PYP is a swollen conformation relative to that of wild-type. This pre-swollen conformation in pG may facilitate the conversion to pB2, resulting in the modest acceleration of the pB1 → pB2 transition.”
Minor points:
Response 1: We corrected this typo in the revised version.
Response 2: We corrected this typo in the revised version. In addition, we have added a new figure (Figure S1) for clear presentation of the crosslink position.
Figure S1. The 7th (glycine) and 91st (methionine) of PYP were mutated to cysteines. The residues are shown in red.
In addition, we have changed the following sentence in the main text.
“only the mutant residues at the 7th and 91st positions were successfully conjugated with the BM(PEG)2 linker.”
to a new sentence
“only the mutant residues at the 7th (glycine) and 91st (methionine) positions were successfully conjugated with the BM(PEG)2 linker (see Figure S1).”
Response 3: We appreciate careful reading. We have corrected this typo in the revised version.
Response 4: We appreciate careful reading. We have corrected this typo in the revised version.
Response 5: We have corrected Figure S2 as the reviewer suggested.
Figure S2. UV-vis spectra of wild-type PYP (black) and BM(PEG)2-incorporated PYP (SML-PYP, red).
Response 6: We appreciate careful reading. We have added the reference section in Supplementary material.

Reviewer 2 Report
The authors have submitted a nice article about a control of conformational changes of photoactive yellow protein (PYP). The rationale behind this study is to understand and control conformational changes in proteins with the aim to utilize them in nanomachines. They introduced a small molecular linker between N- and C-terminal domains of PYP and thus prevented the protrusion of the N-terminus upon light absorption. The effect was further studied by transient absorption spectroscopy, small-angle X-ray scattering and molecular dynamics.
The manuscript is well written without factual and typographical errors. English language is very good and the text is logic. All conclusions are supported by experimental data and the authors avoided useless speculations. The choice of experimental methods is appropriate to the problems solved. I have managed to find only two tiny errors:
l. 45: photorespomsive should be replace by replaced by photoresponsive
l. 233: S,ML-PYP - SML-PYP
Author Response
Response to Reviewer 2 Comments
Comments and Suggestions for Authors:
The authors have submitted a nice article about a control of conformational changes of photoactive yellow protein (PYP). The rationale behind this study is to understand and control conformational changes in proteins with the aim to utilize them in nanomachines. They introduced a small molecular linker between N- and C-terminal domains of PYP and thus prevented the protrusion of the N-terminus upon light absorption. The effect was further studied by transient absorption spectroscopy, small-angle X-ray scattering and molecular dynamics.
Response 1: We appreciate careful reading of our manuscript and comments. We corrected this typo in the revised version.
Response 2: We corrected this typo in the revised version.
Reviewer 3 Report
The authors aim at addressing a highly ambitious general question, which is how to finely control protein motions by protein engineering. They chose a model they have already worked on in the past, the photoactive yellow protein (PYP). They added an artificial linker between the N-terminal domain and the core domain of the protein to constrain its structure and characterized this SML-PYP protein under blue light irradiation by transient absorption spectroscopy and SAXS.
Although the idea is interesting, this manuscript suffers from i/ the choice of the model (SML-PYP) and ii/ the choice of the structural method used to analyse conformational changes in SML-PYP (SAXS is a low resolution technique that will not enable investigating fine structural changes).
The weakness of PYP as a model for nanomachines resides in the fact that structural changes occurring during the photocycle are as much due to partial unfolding as to domain motions. SAXS data must thus be analyzed by using an ensemble of structures and cannot be interpreted with a single structure. In this respect rigid body simulations to refine and filter out an average structure are not adapted here and yield a biased view of a structure that is far less well less defined than the presented models.
Moreover, the SAXS data indicate that the ground state of SML-PYP resembles the light state of SMS-PYP, which itself seems to be similar to the light state of WT-PYP. This raises the question of the structure of the ground state of SMS-PYP. Given that the structure of the light state of WT-PYP (previously calculated by the authors from sparse ESR/NMR constraints) seems to contain a high degree of disorder, it must be inferred that addition of the linker on the protein destabilizes the ground state (which would be compatible with lower time-constants observed for transitions). The authors should therefore have tried to obtain a high resolution structure of SML-PYP, either by crystallizing SML-PYP or by NMR, to assess the ground state structure. If this was not possible they should have checked the perturbations induced by the Cys mutations and subsequent addition of the linker by NMR, as they did it in a previous publication. They should have at least checked the structural integrity of the mutated protein with and without linker by a method like circular dichroism and compared it to WT-PYP.
More details should have been given for the engineered protein. The sites of the Cys mutations are not explicitly indicated (7th and 91th amino acid types must be indicated). This would give an indication about the potential perturbation induced by replacing a side chain involved in important contacts by another. No experimental detail is given for refining of the structures by MD simulations.
In conclusion, in the absence of any high resolution structural data, the authors cannot conclude on the nature of conformational changes that possibly take place, except the obvious fact that they constrained intra-domain motions by cross-linking. In particular the slight changes in overall size observed by SAXS can be ascribed to unfolding, which was not taking into account by the authors when they generated structural models. Unless the authors discuss this aspect and tune down their conclusions, this paper cannot be published in the present form.
Author Response
Response to Reviewer 3 Comments
Comments and Suggestions for Authors:
The authors aim at addressing a highly ambitious general question, which is how to finely control protein motions by protein engineering. They chose a model they have already worked on in the past, the photoactive yellow protein (PYP). They added an artificial linker between the N-terminal domain and the core domain of the protein to constrain its structure and characterized this SML-PYP protein under blue light irradiation by transient absorption spectroscopy and SAXS. Although the idea is interesting, this manuscript suffers from i/ the choice of the model (SML-PYP) and ii/ the choice of the structural method used to analyse conformational changes in SML-PYP (SAXS is a low resolution technique that will not enable investigating fine structural changes).
Response 1: We appreciate the critical comment. As the referee pointed out, in principle the light-adapted species observed by the SAXS experiment should be explained by the ensemble averaged structure, not an optimized protein structure determined from the rigid-body modelling. Judging from the good agreement of the dummy-atom modelled shape and the rigid-body modelled structure shown in Fig. 4, we tentatively argue that the atomistically modelled structure is a protein conformation of the microstate included in the structural ensemble.
To avoid any overinterpretation for the modelled protein structures, we added the following sentences to the “discussion” section.
“We note that because the experiment-restrained rigid-body MD simulation yields only a single protein structure, the resulting structure cannot properly represent the ensemble with the conformational heterogeneity. Nonetheless, judging from the good agreement of the dummy-atom modelled shape and the structure from the experiment-restrained rigid-body MD simulation shown in Figure 4, we suggest that the latter may represent a protein conformation of the microstate included in the ensemble.”
Response 2: We would like to thank the reviewer for the careful and constructive review. We agree that SML-PYP is less stable than the wild-type PYP. This can be already seen by the much increased Rg value. We feel that the crystallography and NMR studies are beyond the scope of the current manuscript, but following the reviewer's comment, we have measured and CD spectra of PYP, mutated PYP and SML-PYP (Figure S5). The CD spectrum of the mutant PYP (G7C-M91C PYP) is quite similar to that of wild-type PYP. On the other hand, the CD spectrum of SML-PYP indicates that its a-helix content is moderately reduced. Considering that the environment around the chromophore region is well conserved similar with wild-type (see Figure S2), the instability may be originated from the conformational perturbation in the vicinity of N-terminals linked with the incorporation of small molecule linkage.
Figure S5 is in the attachment.
Figure S5. Circular dichroism spectra of wild-type PYP, G7C-M91C mutant, and SML-PYP. Whereas wild-type PYP and mutated PYP show similar CD spectra, the CD spectrum of SML-PYP indicates that its a-helix content is reduced probably due to unfolded N-term region.
In addition we have added a new figure (Figure S5) and the following sentence in the Materials and Methods.
“CD spectra were also measured to check structural stability (Figure S5).”
Also we have added the following sentence in the discussion section.
“The CD spectrum of the mutant PYP (G7C-M91C PYP) is quite similar to that of wild-type PYP, whereas the CD spectrum of SML-PYP indicates that its a-helix content is moderately reduced (Figure S5).”
“The reduced a-helix content, the increased Rg value and the well-conserved chromophore environment all together suggest that SML-PYP is swollen conformation with well-conserved core.”
Also we have added the following paragraph in the Supplementary Material.
“Circular Dichroism. To confirm the secondary structure of the G7C-M91C mutant PYP and SML-PYP, circular dichroism (CD) spectra were measured using a CD spectrometer (Jasco-815, JASCO Inc., Japan) with a 2-mm quartz cuvette at room temperature. The spectral window ranged from 190 nm to 260 nm at 0.2-nm intervals. The baseline was measured with the same buffer in the same cuvette and subtracted.”
Response 3: We appreciate this valuable comment. In the revised version, we added a new figure (Figure S1) for clear presentation of the crosslink position. The side chains of 7th and 91st residues do not interfere with any other residues, and therefore replacing these residues with cysteine is not expected to significantly perturb the protein structure.
Figure S1 is in the attachment.
Figure S1. The 7th (glycine) and 91st (methionine) of PYP were mutated to cysteines. The residues are shown in red.
In addition, we have changed the following sentence
“only the mutant residues at the 7th and 91st positions were successfully conjugated with the BM(PEG)2 linker.”
to a new sentence
“only the mutant residues at the 7th (glycine) and 91st (methionine) positions were successfully conjugated with the BM(PEG)2 linker (see Figure S1).”
Also we have added the following paragraph in the main text.
“When we chose the two positions for cysteine mutations where the small molecule linker can be conjugated, we considered the following points. Since our goal is to restrict the photoinduced N-terminus movement of PYP, one of the positions should be located at the N-terminus. For this position, we chose the 7th (glycine) and 13th (Asparagine) positions considering the minimal interference with other residues and the side chain direction pointing toward the protein body. For the other position in the protein body, we chose 91st (methionine) and 113st (leucine) positions considering the minimal interference and the side chain direction pointing toward the N-terminus. Another consideration at this point was that the selected position at the N-terminus and the position at the protein body should be separated by a distance similar to or shorter than the length of the small molecule linker.”
Response 4: The original manuscript already has explanations on the experiment-restrained rigid body MD simulations, but in the revised manuscript, we further strengthened the details
Response 5: We agree with this comment. Our conclusion in the previous manuscript were overclaimed. As the reviewer suggests, in the revised version, we have tried to tune down the conclusion. In particular, the following sentence
“SAXS and TA results show that cross-linking between the N-terminus and the core of PYP with BM(PEG)2 blocks the N-terminal movement of PYP and increases the size of the protein.”
has been replaced with the following new sentence.
“SAXS and TA results show that cross-linking between the N-terminus and the core of PYP with BM(PEG)2 may constrain the N-terminal movement of PYP.”
Also the following sentence
“The ability to regulate the mechanical motions with cross-linking will open a new strategy for the mechanical manipulation of protein nanotechnology and lead to novel nanomachines for a variety of applications.”
has been replaced with the following new sentence.
“The ability to regulate the mechanical motions with cross-linking could provide a new strategy for the mechanical manipulation of protein nanotechnology and lead to novel nanomachines for a variety of applications. “
In addition, we have changed the discussion section in main text.
“The CD spectrum of the mutant PYP (G7C-M91C PYP) is quite similar to that of wild-type PYP, whereas the CD spectrum of SML-PYP indicates that its a-helix content is moderately reduced (Figure S5). The SAXS data show the increased of Rg value in SML-PYP by 0.9 Å, which corresponds to an approximate volume increase of 5,500 Å3. The volume of the 6 residues in the N-terminal region would be about 147 Å3 plus the volume of the hydration shell and thus this alone cannot account for the increase of Rg value by 0.9 Å. On the other hand, the environment around the chromophore region of SML-PYP is expected to be well conserved and similar with that of wild-type on the basis of the similarity of UV-vis spectra (see Figure S2). The reduced a-helix content, the increased Rg value and the well-conserved chromophore environment all together suggest that SML-PYP is swollen conformation with well-conserved core. In accordance with this possibility, the molecular shape reconstructed from the SAXS data shows the partial protrusion in the ground state of SML-PYP (Figure 4a). The molecular origin of the partial protrusion can be provided by comparing the shape and the structure from the experiment-restrained rigid-body MD simulation (Figure 4d). The protrusion of the reconstructed shape well mataches the N-terminal region in the structure from the experiment-restrained rigid-body MD simulation, confirming that the N-terminal region is indeed responsible for the protrusion. This result, which we did not anticipate when we had chosen the cross-linking positions and the small molecule linker, shows that the overall shape of the protein molecule can be easily affected by the small molecule linker. We note that because the experiment-restrained rigid-body MD simulation yields only a single protein structure, the resulting structure cannot properly represent the ensemble with the conformational heterogeneity. Nonetheless, judging from the good agreement of the dummy-atom modelled shape and the structure from the experiment-restrained rigid-body MD simulation shown in Figure 4, we suggest that the latter may represent a protein conformation of the microstate included in the ensemble.”

Round 2
Reviewer 1 Report
The manuscript was significantly improved, and I found no necessity of further revision. Now I recommend the publication in present form.
Reviewer 3 Report
The manuscript has greatly improved since the original submission. A number of data has been added, which yields a more precise description of SML-PYP in its various states. The authors have satisfyingly answered all concerns addressed in Report 1. Only minor corrections now need to be carried out, as listed below:
line 68: protrusion of N-terminus -> protrusion of the N-terminus
line 107: Asparagine -> asparagine
line242: serve the starting point -> serve as the starting point
line246: the increased of Rg -> an increase of Rg
line 250: similar with -> similar to
lines 252-53: SML-PYP is swollen conformation with well-conserved core -> SML-PYP is in a swollen conformation with a well-conserved core
line 258: mataches -> matches
line 348: we measure UV-vis spectrum -> we measured a UV-vis spectrum
Also, in Figure S1, please indicate the PDB accession number for the visualized structure (presumably 2PHY?)
